# Monoclonal Antibodies and Airway Diseases

**DOI:** 10.3390/ijms21249477

**Published:** 2020-12-13

**Authors:** Annina Lyly, Anu Laulajainen-Hongisto, Philippe Gevaert, Paula Kauppi, Sanna Toppila-Salmi

**Affiliations:** 1Inflammation Centre, Skin and Allergy Hospital, Helsinki University Hospital, University of Helsinki, P.O. Box 160, 00029 HUS Helsinki, Finland; sanna.salmi@helsinki.fi; 2Department of Otorhinolaryngology—Head and Neck Surgery, Helsinki University Hospital, University of Helsinki, 00029 HUS Helsinki, Finland; anu.laulajainen-hongisto@hus.fi; 3Department of Otorhinolaryngology, Upper Airway Research Laboratory, Ghent University Hospital, 9000 Ghent, Belgium; philippe.gevaert@ugent.be; 4Heart and Lung Center, Pulmonary Department, University of Helsinki and Helsinki University Hospital, 00029 HUS Helsinki, Finland; paula.kauppi@hus.fi; 5Medicum, Haartman Institute, University of Helsinki, 00029 HUS Helsinki, Finland

**Keywords:** airways, asthma, chronic rhinosinusitis, biologicals, monoclonal antibody

## Abstract

Monoclonal antibodies, biologics, are a relatively new treatment option for severe chronic airway diseases, asthma, allergic rhinitis, and chronic rhinosinusitis (CRS). In this review, we focus on the physiological and pathomechanisms of monoclonal antibodies, and we present recent study results regarding their use as a therapeutic option against severe airway diseases. Airway mucosa acts as a relative barrier, modulating antigenic stimulation and responding to environmental pathogen exposure with a specific, self-limited response. In severe asthma and/or CRS, genome–environmental interactions lead to dysbiosis, aggravated inflammation, and disease. In healthy conditions, single or combined type 1, 2, and 3 immunological response pathways are invoked, generating cytokine, chemokine, innate cellular and T helper (Th) responses to eliminate viruses, helminths, and extracellular bacteria/fungi, correspondingly. Although the pathomechanisms are not fully known, the majority of severe airway diseases are related to type 2 high inflammation. Type 2 cytokines interleukins (IL) 4, 5, and 13, are orchestrated by innate lymphoid cell (ILC) and Th subsets leading to eosinophilia, immunoglobulin E (IgE) responses, and permanently impaired airway damage. Monoclonal antibodies can bind or block key parts of these inflammatory pathways, resulting in less inflammation and improved disease control.

## 1. Introduction

Chronic inflammatory airway diseases include several overlapping morbidities, such as asthma and chronic obstructive pulmonary disease (COPD) in the lower airways; and allergic rhinitis (AR), nonallergic rhinitis (NAR), and chronic rhinosinusitis (CRS) in the upper airways. AR has a prevalence of 20–30%, NAR has a prevalence of 10%, and CRS has a prevalence of 10–20%, and these common diseases cause remarkable suffering and costs [1,2,3]. They can be subdivided based on such as age of onset, presence of allergy (skin prick test or systemic allergen specific immunoglobulin E (IgE)), with or without nasal polyps and/or T helper (Th) cell 2 prominent inflammation. Exposure to environmental irritants (such as smoking and occupational exposure), recurrent infections, lifestyle factors (such as obesity, stress), co-existing diseases, and genetic/epigenetic factors play a role in disease onset and progression [4,5]. The diagnostic methods include clinical examination, lung function tests, allergy tests, and paranasal sinus computed tomography scans [5,6,7]. Symptom control of mild cases can be well achieved by the basic treatment such as inhaled/intranasal corticosteroids, inhaled beta agonists, antihistamines, and nasal lavage [5,6]. Patients with moderate to severe forms often suffer from recurrent infective exacerbations and disease recurrence/progression despite maximal baseline therapy and surgeries. Hence, they require advanced diagnostic methods and therapeutics. Antibodies are an important part of humoral adaptive immunity and homeostasis. They also play a role in airway diseases such as IgE in allergy and CRS with nasal polyps (CRSwNP), antibody deficiency in CRS, and aberrant antiviral IgG responses in asthma exacerbations [5,8]. Since their introduction about five decades ago, a wide range of monoclonal antibodies are nowadays commercially available and have been largely used in basic and clinical science of airways. This review focuses on presenting two main airway pathologies of human adults: asthma and CRS. We first introduce monoclonal antibodies and their role in biomarker diagnostics of adult asthma and CRS. Secondly, we present the role of monoclonal antibodies as advanced therapeutics of asthma/CRS.

## 2. Monoclonal Antibodies

Antibodies (immunoglobulin (Ig) A, IgD, IgE, IgG, IgM) are secreted by B-cells that are activated to plasma cells after antigen presentation in regional lymph nodes or secondary lymphoid organs (Figure 1) [9]. Monoclonal antibodies (mAbs) come from a single B-cell parent clone and recognize specifically a single epitope per antigen [10]. Antibodies are crucial to make leukocytes (such as T killer cells) to detect and destroy pathogens and infected host cells. MAbs are made for laboratory and therapeutic use by various techniques. The first technique described in 1975 was based on creating a hybridoma by combining an activated B-cell from an immunized animal spleen and immortalized myeloma cell, resulting in a stable hybrid cell line producing monoclonal antibody [11]. The first mAbs used in therapeutic purposes were of murine origin, which generated unwanted immunogenic reactions and human anti-mouse antibody formation [12]. The revolution of molecular biology techniques has enabled the production of humanized and fully human mAbs that have helped to tackle this problem, although anti-drug antibodies are still one of the outcomes of immunogenicity [12]. For research and laboratory use, there are exponential numbers of commercially available specific monoclonal antibodies for immunoassays such as immunohistochemistry, immunofluorescence and enzyme-linked immunosorbent assay (ELISA) [13]. Since their invention about 50 years ago, there has been a large interest to use monoclonal antibodies in experiments to discover relevant proteins and pathways behind airway pathologies [14,15].

## 3. Chronic Inflammatory Airway Diseases

### 3.1. Asthma and Airway Allergy

Asthma is characterized by reversible lower airway obstruction [6]. Reversible obstruction can be resolved spontaneously by time but also by bronchodilators. Typical asthma symptoms include recurrent or prolonged (over 8 weeks) cough, wheezing, dyspnea, nighttime symptoms, and an overproduction of mucus. Reversible airway obstruction can be diagnosed by spirometry and peak expiratory flow (PEF) monitoring, and bronchial hyperresponsiveness is detected with methacholine or mannitol challenge or by exercise test [6,16,17]. This reversible obstruction is caused by inflammation of bronchus epithelia and is associated with Th2 type inflammation and cytokines in approximately 50% of adult asthma patients. The prevalence of asthma varies, being up to 10% in adults [16]. Severe asthma has been defined as needing drug therapy at Global Initiative for Asthma (GINA) step level 4 or 5 and having recurrent per oral corticosteroid courses or maintenance treatment with per oral corticosteroid or having recurrent exacerbations or at least one severe exacerbation in the last 12 months [6]. Severe uncontrolled asthma is reported in 2.3–3.6% of patients with chronic asthma. Severe uncontrolled eosinophilic asthma has been estimated to form less than 1% of all asthma [18].

### 3.2. CRS

CRS is defined as the presence of two or more symptoms, one of which is either nasal obstruction or nasal discharge, together with facial pain/pressure or loss of smell, for more than 12 weeks [5]. The overall prevalence of CRS has been estimated to be 10.9%, with wide variation between countries (6.9% to 27.1%) [2]. Traditionally, CRS has been classified into two subtypes: CRS with nasal polyps (CRSwNP) and without nasal polyps (CRSsNP), which is diagnosed after endoscopic evaluation of the presence of bilateral polyps in the middle meatus. Data on the overall prevalence of CRSwNP are limited, but it is estimated to be approximately 2–3% [5,19,20]. However, the inflammatory profile of CRS has proven to be more complex than whether or not polyps are present [21]. Therefore, the new European Position Paper on Rhinosinusitis and Nasal polyps (EPOS) guidelines propose a new clinical classification, which is based first on the etiology (primary vs. secondary) and then the localization of the disease (unilateral vs. bilateral), followed by the evidence of either type 2 or non-type 2 inflammation [5]. The diagnostics of CRS consist of clinical examination including nasal endoscopy, computed tomography imaging, validated patient symptom questionnaires (for example, the sinonasal outcome test SNOT-22), olfactory tests, and histopathologic examination of inflamed tissue [5]. The symptoms are caused by chronic inflammation of the upper airway mucosa, and the prolonged inflammation may cause tissue remodeling [5]. In CRS, the remodeling of sinonasal tissues consists mostly of polyp formation, goblet cell hyperplasia, and epithelial barrier abnormalities. Barrier remodeling results in greater permeability, facilitating prolonged or recurrent CRS [5]. All of these changes are usually seen in type 2 CRS and, despite being only a minority of the cases, the most severe forms of CRS leading to high use of oral corticosteroids/antibiotics and recurrent surgeries are often type 2 CRS [5]. To evaluate the endotype, the combination of phenotype (e.g., CRSwNP, non-steroidal anti-inflammatory drugs (NSAID) exacerbated respiratory disease (N-ERD), co-morbid asthma), response to treatment (surgery, systemic corticosteroids, antibiotics) and also markers such as polyp eosinophilia are key instruments to estimate it [5,22,23].

### 3.3. Co-Morbid Asthma and CRS

Asthma, CRS, and AR are all multifactorial chronic airway diseases that have partly overlapping pathogenetic mechanisms and risk factors [24]. These environmental risk factors, such as exposure to pollution and tobacco smoke, are linked to asthma and CRS pathogenesis and disease aggravation via the disruption of interplay of epithelial barriers with particles, allergens, and microbes [25,26]. In CRS, the most commonly discussed microbial agent is *Staphylococcus aureus*, but some evidence also implicates dysbiosis of the microbial community as a whole rather than a specific dominant pathogen [5,27].

CRS, AR, and asthma all have several subforms. The main phenotypes of CRS are CRSsNP and CRSwNP [5], yet there are additional subtypes of CRS such as allergic fungal rhinosinusitis, isolated sinusitis, eosinophilic CRS, central compartment allergic disease, and non-eosinophilic CRS. In AR, there is evidence that the sensitization profile and/or allergic multimorbidity are associated with morbidity in children [28] and adults [29]. In asthma, differences are seen between different asthma types in their genetic backgrounds, association with AR and CRS, and possibly also in microbe–host interactions [26]. Childhood-onset asthma is, more often than adult-onset asthma, associated with genetic predisposition, whereas the background of adult-onset asthma is often multifactorial. In adult-onset asthma, the activation of inflammatory molecular pathways leads to persistent mucosal inflammation, variable airway obstruction, and tissue remodeling [30].

Up to 40% of patients with CRSwNP and asthma are hypersensitive to acetylsalicylic acid (ASA) and/or other non-steroidal anti-inflammatory drugs (NSAID) [31]. NSAID exacerbated respiratory disease (N-ERD) usually includes a triad of CRSwNP, asthma, and hypersensitivity to ASA and/or other NSAIDs. Abnormalities of the cyclooxygenase (COX) pathway, severe eosinophilic hyperplastic inflammation, and tissue remodeling with fibrosis in both paranasal sinuses and lower airways are characteristics of N-ERD [32,33,34,35]. The treatment of N-ERD consists of conventional asthma and CRS medications; however, repeated oral corticosteroid courses are often also needed [36,37]. The CRSwNP of N-ERD patients is often resistant to medical treatments and may lead to repeated paranasal surgeries.

Oral ASA treatment after desensitization (ATAD) has shown to be effective in improving quality of life (QOL) and total nasal symptom score in patients with N-ERD [5,38]. However, the treatment is associated with adverse effects (typically gastrointestinal) and should be continued strictly on a daily basis [5]. Studies with ATAD show high discontinuation rates, and not all patients benefit from it [5,39]. Monoclonal antibodies have shown efficacy in patients with severe CRS + N-ERD [40,41,42].

### 3.4. Mechanisms behind Airway Diseases

The airway epithelium secretes cytokines such as thymic stromal lymphopoietin (TSLP), interleukin 33 (IL-33), and IL-25 in response to tissue damage, pollutants, pathogen recognition, or allergen exposure [43] (Figure 1). These cytokines are involved in the activation of innate lymphoid cells type 2 (ILC2). The activation of ILC2 leads to the release of IL-9, IL-4, IL-13, and IL-5, and to Th2 type inflammation both in asthma and CRS [44,45,46]. IL-9 has a role in mast cell involvement and airway hyperreactivity [47]. IL-4 and IL-13 are involved in B-cell maturation and IgE production [43]. IL-5 is a growth factor important for the proliferation, maturation, and activation of eosinophils, which associate both with asthmatic inflammation and with (IgE-dependent) allergic inflammation. Eosinophilia, IgE production, and goblet cell hyperplasia result from the Th2 type cytokines [48,49]. Eosinophilic inflammation is found both in allergic and non-allergic conditions. Allergen sensitization is developed when naïve T lymphocytes are differentiated into Th2 type cells and further allergen-specific IgE producing B-cells [7].

CRS is often associated with mucociliary dysfunction [50,51]. Microbial agents, especially *S. aureus*, and microbiome dysbiosis seem to play an important role in CRS pathogenesis. *S. aureus* can directly affect mucosal barrier function leading to type 2 inflammation via serine protease-like protein (Spl), TSPL and IL-33 [52,53]. *S. aureus* colonization is especially common in patients with CRSwNP, but *S. aureus*-specific IgE has been associated with both CRSwNP and asthma [22,52,53]. Type 2 cytokines inhibit t-PA (tissue plasminogen activator), which results in fibrin mesh deposition to form the nasal polyp tissue matrix [54].

## 4. The Role of Antibodies in Airway Diseases

Allergic asthma/AR/allergic conjunctivitis are characterized by a type 2 dominated immune response associated with increased serum IgE levels in response to inhaled allergen. Specific IgE for several allergens has been shown to be a risk factor of asthma in children and young adults [55]. There is also evidence that AR is a predisposing factor of adult-onset of asthma [29]. There is no clear evidence of an association between airway allergy and CRS [5].

### 4.1. IgE

Specific IgE to allergens and pathogenes (such as *S. aureus* superantigens) have been demonstrated in the nasal polyp tissue of CRSwNP patients as well as in the bronchial tissue of asthma patients [56,57,58]. Local IgE production might play a role in CRSwNP pathogenesis and polyp regrowth after sinus surgery [59]. The mechanisms by which *S. aureus* leads to type 2 cytokine signaling in airways is not fully understood. *S. aureus* colonization might benefit from type 2 inflammation, as it may suppress normal immune responses against it. *S. aureus* secretes superantigenic toxins that modify host immune responses toward the production of local polyclonal IgE [22]. It has been suggested that type 2 inflammation is triggered by *S. aureus* via Toll-like receptor 2, which is a pattern recognition receptor [60,61,62]. Studies have also been shown that the Asian CRSwNP population has more type 2 low inflammation, and CRSwNP tissue of Asian patients present lower superantigen effects [63,64].

### 4.2. Other Antibodies

Other antibodies have been less studied in asthma/CRS. Primary antibody deficiencies have been shown to be related to CRS, such as common variable immune deficiency (CVID), selective IgA deficiency, IgG subclass deficiency, and specific antibody deficiency [65]. Airway infections are very frequently involved in triggering asthma or CRS exacerbations [5]. The anti-virus effects of monoclonal antibodies are indirect—for example, improvement of the antiviral capacity of dendritic cells [66]. Antiviral mAb therapy usually directly and rapidly targets the infectious agent, yet evidence has revealed that antiviral mAbs may be used to recruit the endogenous immune systems of infected organisms to induce long-lasting vaccine-like effects [67]. A study of French adult asthmatics has shown that patients hospitalized for asthma exacerbations with a positive virus sample had lower serum IgG level than did their virus negative counterparts [68]. Moreover, longer hospital stays and longer duration of oral steroids were linked to lower serum IgG concentrations, suggesting that severe exacerbations could be related to aberrant antiviral IgG production; however, more studies are needed to confirm this.

## 5. Monoclonal Antibodies and Diagnostics of Airway Diseases

Monoclonal antibodies are widely used for research purposes of airway diseases and are in clinical use in allergy diagnostics. Blood tests remain an important component in asthma diagnostics; the detection of elevated IgE levels and eosinophils can be used to help identify allergen sensitivity.

### 5.1. Measurement of Total and Specific IgE in Airway Diseases

Specific IgE (i.e., IgE directed against a specific allergen) and eosinophil counts were confirmed as the most consistent biomarkers to measure the risk of asthma in children [69]. The measurement of specific and total serum IgE levels can be useful to distinguish between allergic and non-allergic asthma, although reports suggest that about 30% of asthmatic patients with a negative skin prick test results have high total circulating IgE (>150 U/mL) [8]. If specific IgEs are considered for selecting an appropriate biologic agent, screening for perennial allergens such as dust mite would have the best rationale [69].

### 5.2. Potential Biomarkers for Airway Diseases

In order to predict outcomes and therapeutic responses of asthma and/or CRS, there is active research on biomarkers, such as type 1, 2, and 3-related cytokines. Type 1, 2, and 3 responses are evoked by natural mucosal immune responses against viruses/bacteria, parasites, and bacteria/fungi correspondingly, and they are characterized by certain cytokine profiles: IFN-γ and IL-12 in type 1, IL-4, IL-5, and IL-13 in type 2, and IL-17A and IL-22 in type 3 [5]. In pathogen penetration, single or combined type 1, 2, and 3 immunological responses are invoked to eliminate the pathogen, whereas in asthma and CRS, aberrant type 2 responses (and to a lesser extent also type 1 and 3) play a role in airway disease severity and therapeutic responses [70,71].

Patients selected by biomarkers might obtain a greater benefit from therapy with anti-IL-13 mAbs [72]. Elevated bronchial expression of IL-5 and IL-13 has been shown to be associated with sputum and blood eosinophilia and moderate-to-severe asthma [72]. IL-13 and IL-4 regulate the synthesis of IgE and are thus important biomarkers of Th2 cell activation. As a result of IgE binding with the high-affinity receptor (FcεRI) found in basophils and mast cells, there is a cellular activation that ends in the liberation of various inflammatory mediators including cytokines such as IL-5, IL-4, and IL-13. IL-13 and IL-4 also induce periostin. Periostin binds to several integrin molecules on the epithelial cell surface to support the adhesion and migration of epithelial cells, and elevated airway mucosal periostin may be useful in detecting type 2 CRSwNP [73,74,75] and asthma [76]. Automated immunoassays have been shown to be a potent test for measuring human serum IL-13 [77] and periostin [78] concentrations for clinical purposes. De Schryver et al. have shown that methylprednisolone and omalizumab significantly reduce serum periostin levels and that the periostin expression is in accordance with clinical outcome [79]. However, serum levels of biomarkers are not specific for asthma or CRS. Elevated periostin levels have been detected for example in pulmonary fibrosis and lung carcinoma, and elevated type 2 cytokines or serum IgE in helminth infections and atopic dermatitis [27,80].

## 6. Monoclonal Antibodies and Treatment of Airway Diseases

Unraveling the pathogenesis of diseases has provided the basis for the pharmaceutical industry to develop protein drugs, or “biologics”, with higher specificity and mechanism of action than small molecule drugs. In 2015, monoclonal antibodies were the most important class of biologics approved by the United States Food and Drug Administration (FDA) [81], and their utilization in therapy has rapidly increased since. Personalized medicine is addressing the issue of providing targeted treatment for the right patient [82]. The endotype-driven treatment approach requires careful selection of the patient population who might benefit from a treatment by advanced therapies [83,84]. In the following chapter, mAbs used to treat asthma and CRSwNP are introduced; their main mechanisms of actions are illustrated in Figure 1.

### 6.1. Commercially Available Monoclonal Antibodies and Their Mechanisms of Action

#### 6.1.1. Omalizumab—Anti-IgE

In the fast phase of allergic reaction, allergen-specific IgE produced by B-cells binds to high affinity FcR (FceRI) expressed on immune cells such as basophils and mast cells. Then, allergen exposure can lead to antigen cross-linking IgE molecules on the same mast cell, receptor aggregation, and initiation of the intracellular signal cascade leading to degranulation and the release of histamine, prostaglandins, and cytokines that mediate the clinical manifestations of atopy [85]. Omalizumab, a humanized IgG1/k monoclonal antibody, targets the Fc region of IgE, and by binding to free IgE in blood and body fluids, it neutralizes the ability of IgE to bind to its receptor (FcεRI, high-affinity receptor and FcεRII, low-affinity receptor) [86]. On top of inhibiting the cross-linking on mast cells, this induces the down-regulation of IgE receptor expression on other immune cells such as basophils and dendritic cells [87,88]. Omalizumab was the first biological therapy developed for asthma, and it has now been used for 15 years. During these years, the functions of IgE in bronchial asthma have proven to be more complex than that of the classical role in allergy and anaphylaxis (reviewed in [89]). For example, smooth muscle cells in lung tissue have receptors for IgE, and it is involved in their proliferation, independent of the presence of allergens. IgE also plays a role in non-allergic diseases such as chronic idiopathic urticaria and CRSwNP and is involved in eosinophilic inflammation [89].

#### 6.1.2. Mepolizumab and Reslizumab—Anti-IL-5

Type 2 inflammation present in asthma and CRSwNP is featured with airway eosinophilic infiltration, particularly in nasal polyps. Eosinophils are also frequently elevated in peripheral blood in type 2 asthma. High eosinophil levels are associated with exacerbations and bronchial obstruction [90]. The key mediator of eosinophils is interleukin-5 (IL5), being responsible for their differentiation, growth, activation, and survival as well as recruitment to airways [91,92]. Mepolizumab is a humanized IgG1/k monoclonal antibody toward IL-5, binding to it with high affinity and preventing its linkage to IL-5Rα [93,94]. Reslizumab is a humanized IgG4/κ monoclonal antibody specifically interacting with the epitope IL-5 uses to bind its receptor IL-5Ra, thereby blocking its bioactivity [95].

#### 6.1.3. Benralizumab—Anti-IL-5Ralpha

Different from mepolizumab and reslizumab, benralizumab binds to IL-5-receptor instead of its ligand. Benralizumab is an afucosylated humanized IgG1/κ monoclonal antibody, selectively recognizing the IL-5Rα subunit [96]. The interaction of benralizumab with IL-5Rα prevents IL-5 binding to target cells and impedes the heterodimerization of IL-5Rα and βc subunits, thus inhibiting the activation of IL-5-dependent signaling cascades. In addition, benralizumab binds to the FcγRIIIa membrane receptor expressed by natural killer cells through the constant Fc region. FcγRIIIa activation induces the eosinophil apoptosis mechanism called antibody-dependent cell-mediated cytotoxicity, which is amplified by afucosylation [97], resulting in depletion of the blood eosinophils. A recent study describes also reduction in the number of basophiles after treatment with benralizumab [98].

#### 6.1.4. Dupilumab—Anti IL-4Ralpha

Dupilumab is a fully human monoclonal antibody to the interleukin-4 receptor α subunit, IL-4Ralpha, which is utilized by two cytokines IL-4 and IL-13 [99]. IL-4 mediates its biological effects by binding to IL-4Rα, which is followed by the recruitment of either gamma c or IL-13 receptor alpha 1 (IL-13Rα1) to form a signaling complex [100]. IL-13 binds to IL-13Rα1 and then forms a signaling complex by recruiting IL-4Rα [100]. Altogether, IL-4Ralpha is involved in three different combinations of receptor complexes, and the intracellular response potencies are varied between the binding ligand, IL-4 vs. IL-13 [100,101].

Due to the shared receptor, IL-4 and IL-13 also have overlapping functions, and these sister cytokines act both cooperatively as well as independently in type 2 inflammation cascades. Both interleukins promote B-cell proliferation and class switch to IgG4 and IgE [102]. IL-13 is a cytokine secreted by activated Th2 cells, and it acts as an important mediator of allergic inflammation pathogenesis. Distinct functions for IL-13 include tissue remodeling, goblet cell mucus hypersecretion, subepithelial fibrosis, and emphysematous changes [103]. IL-4 and IL-13 can both induce Th2 cells and epithelial cells to produce eosinophil-promoting factors (i.e., IL-5 and eotaxins) and stimulate eosinophils to migrate to sites of inflammation from blood [104]. However, a recent murine model study shows that only dual IL-4/IL-13 blockade prevented type 2 inflammation broadly enough to prevent lung-function impairment—blocking only IL-4 or IL-13 alone was not enough to provide major clinical benefits [105]. This has been seen also in clinical experiments with IL-4 and IL-13 blockers for the treatment of type 2 diseases [106]. Dual blockade of IL-4/IL-13 with dupilumab halted eosinophil infiltration into lung tissue in mouse model without affecting circulating eosinophils, demonstrating that tissue, but not circulating eosinophils, contribute to disease pathology [105].

### 6.2. Monoclonal Antibodies in Asthma Treatment

Monoclonal antibodies are considered as a treatment option for severe asthma [107]. First, the patient’s symptoms are carefully assessed in order to estimate if the patient truly has asthma, if the current symptoms are associated with asthma, if the current asthma drug therapy is adequate, if the patient is adherent for the drug therapy, and that there are no environmental factors that should be considered [17,107]. Poor symptom control, frequent yearly exacerbations or serious exacerbations, and diminished lung function are signs of uncontrolled asthma and an indication for biologicals if the situation is not controlled with other maximal medication [107]. Controlled asthma that deteriorates if high-dose inhaled corticosteroids or systemic corticosteroids are tapered is another indication for biologicals [107]. The selection of a suitable drug is based both on allergy (whether the patient has allergic asthma to perennial allergens) but also on eosinophils (whether the patient has high or low blood eosinophils) [108]. Contradictory to biologicals in rheumatic diseases, the biologicals targeting IgE or Th2 cytokines have been well tolerated and safe to use [109,110]. The commercially available antibodies and their therapeutic use is summarized in Table 1.

The first monoclonal antibody treatment for lower airway diseases was anti-IgE therapy with omalizumab. After that, anti-IL5-, anti-IL-5Ralpha- and anti-IL-4R-treatment have been introduced for the treatment of asthma. These monoclonal antibodies are humanized IgG antibodies and selective for their binding capacity. Both anti-IL-5 and anti-IL-5Ralpha-antibody treatment may be associated with anti-drug-antibody development [12].

Omalizumab is indicated as an add-on therapy in adults and children over six years for inadequately controlled severe asthma. Omalizumab reduces asthma exacerbations by OR 0.55 (95% CI 0.42–0.60) and hospitalizations by OR 0.16 (95% CI 0.06–0.42) [111,112]. Furthermore, it is beneficial in the reduction of inhaled corticosteroids [111].

Mepolizumab, reslizumab (anti-IL-5), and benralizumab (anti-IL-5Ra) are used for severe eosinophilic asthma. Mepolizumab reduces exacerbations by approximately 50% in patients with eosinophils at least 150 cells/uL at screening or at least 300 cells/uL in the previous year and high-dose inhaled corticosteroids and at least one additional controller medication [113]. In addition, improvement in QOL, asthma control measures with asthma control questionnaire, and lung function (FEV1) have been reported [113]. Furthermore, mepolizumab has been shown to reduce oral corticosteroid need [113]. Reslizumab reduces asthma exacerbations by OR0.50 (95% CI 0.37–0.67) in patients with medium-to-high dose inhaled corticosteroids and blood eosinophils at least 400 cells per uL and one or more exacerbations in the previous year [114,115].

Benralizumab reduces exacerbations and the need of per oral glucocorticosteroids, and it improves QOL and lung function not only in clinical trials but also in real-world studies [110,116,117]. Benralizumab has been reported to reduce asthma exacerbations from 4.9 to 1.3 per year and to reduce daily prednisolone dose from a median 10 to 0 mg [117].

Dupilumab (anti IL-4Ralpha) reduces asthma exacerbations and improves lung function in patients with moderate to severe asthma [118]. Furthermore, with dupilumab treatment dose of maintenance, oral glucocorticoids can be reduced. Reduction in the oral glucocorticoid dose and elimination of per oral glucocorticoids is more likely in the asthma patients with the baseline level of eosinophils at least 300 cells/mm^3^ [118]. A transient rise in eosinophils is seen more often in the dupilumab-treated patients when compared to placebo-treated patients [118]. In addition, injection-site reactions were more common in the dupilumab-treated patients.

### 6.3. Monoclonal Antibodies in CRS Treatment

Targeted monoclonal antibody therapies have shown encouraging results in the management of severe CRSwNP. As type 2 CRSwNP and asthma largely overlap, also therapeutics are in some cases targeted to both severe asthma and severe CRSwNP. According to the European Position Paper on Rhinosinusitis and Nasal polyps 2020 (EPOS 2020) guidelines, the indications for using biological treatment for CRSwNP include bilateral polyps and at least one previous endoscopic sinus surgery, together with at least three of the following criteria: evidence of type 2 inflammation, need for systemic corticosteroids (or contraindication for it), significantly impaired quality of life, significant loss of smell, or diagnosis of comorbid asthma. The effect of the treatment should be evaluated after 4 months and 1 year, and in case there is no response, treatment should be discontinued [5].

Anti-IgE therapy (omalizumab) is the second and latest biologic therapy approved for CRSwNP by the European Medicines Agency (EMA) in August 2020, and it is pending FDA approval for CRSwNP [41]. A study by Gevaert et al. has shown a decrease of symptom score for nasal congestion, anterior rhinorrhoea, loss of sense of smell, wheeze and dyspnea, and a significant reduction of endoscopic nasal polyp score, radiologic Lund–MacKay score, and asthma symptoms [119]. Another randomized controlled trial (RCT) by Pinto et al. showed improvement in symptoms, but no significant improvement in Lund–Mackay score or other endpoints [120]. In a recent study on patients with N-ERD, both nasal and lung symptoms improved significantly with omalizumab treatment [121]. However, these studies were small, with only around 20 patients in each group. Recent results from two bigger phase 3 RCTs of 265 patients has shown that omalizumab significantly reduced endoscopic nasal polyp score, nasal congestion score, and SNOT-22 score compared to placebo at week 24 [122]. Patients with comorbid asthma reported significant improvement in Asthma Quality of Life Questionnaire scores [122].

Anti-IL-5 treatment with reslizumab was found to decrease nasal polyp scores in an RCT, in which patients received a single injection of reslizumab (*n* = 16) or placebo (*n* = 8) [123]. Studies with mepolizumab have shown a significant reduction in patients’ need for surgery and an improvement in symptoms [5,124,125]. A Cochrane review summarized that mepolizumab may improve both disease-specific and generic health-related quality of life (HRQL), yet its effect to reduce surgery or improve nasal polyp score is uncertain [126]. At the moment, phase 3 RCTs are ongoing for both benralizumab and mepolizumab, altogether with over 800 patients with severe CRSwNP [41]. More information about the efficacy of anti-IL-5-treatments will be available after they are finished. However, preliminary results of RCT of 407 patients has shown that mepolizumab significantly reduced endoscopic nasal polyp score, nasal obstruction VAS score, VAS (overall, composite, loss of smell), SNOT22 score, and the need for surgery [127]. Nasopharyngitis was the most common adverse event in this study (23–25%) [127].

An anti-IL-4/IL-13 drug, dupilumab, is the first monoclonal antibody approved for the treatment of CRSwNP in 2019 [5]. Before that, it has been used for the treatment of asthma since 2018 and atopic dermatitis since 2017. In a double-blind RCT (DBRCT) with 276 patients with severe CRSwNP using regular topical nasal steroids, dupilumab reduced polyp size, sinus opacification, and severity of symptoms (nasal congestion and obstruction, sense of smell) compared with placebo. It also diminished the need for rescue treatment with systemic corticosteroids and sinus surgery [128]. A Cochrane review summarizes that dupilumab has been shown to improve disease-specific HRQL compared to placebo, and it might improve symptoms and generic HRQL and reduce the need for further surgery [126]. Moreover, there is no evidence of an increased risk of serious adverse events; however, there may be little or no difference in the risk of nasopharyngitis [126]. Among dupilumab-treated atopic dermatitis patients, conjunctivitis is the most common side effect [129]. However, in patients with asthma and CRSwNP, the incidence of conjunctivitis was very low, similar as for placebo [129].

### 6.4. Future Monoclonal Antibody Treatments for Airway Diseases

#### 6.4.1. Anti-TSLP

Thymic stromal lymphopoietin (TSLP) is produced by fibroblasts and epithelium and plays a role in T cell maturation. TSLP enhances IL type2 cytokine production in mast cells and activates ILC2s together with IL-33 or IL-25. TSLP has shown to associate with asthma and CRSwNP after virus challenge [130]. Tezepelumab (AMG-157/MEDI9929) is a human anti-TSLP antibody. A DBRCT of 31 mild asthmatics has shown that AMG-157 attenuated allergen-induced early and late asthmatic responses, and it decreased blood and sputum eosinophils [131]. Anti-OX40L promotes regulatory T (Treg) cells and suppresses T-cell mediated inflammation, and hence, it might be a potential therapeutic target for severe asthma [132]. Yet, in a study that used a combination of anti-OX40L and anti-TSLP, the expected effects on Treg-mediated inflammation was not observed [133]. Tezepelumab (anti-TSLP) decreases exacerbations and improves lung function measured by FEV1 (forced expiratory volume in one second) statistically significantly compared to placebo in patients with medium-to-high dose inhaled corticosteroids and long-acting beta-2-agonist [134]. The exacerbation rates were 61–71% lower than in the placebo group depending on the dose of the tezepelumab [134]. A reduction in asthma exacerbations was found irrespective of eosinophil level.

#### 6.4.2. Anti-TNF

Type 2 low pathways might also comprise future targets for monoclonal antibody therapy [135]. Anti-TNF could have potential in patients with neutrophilic non-infectious COPD [136] and in severe asthma with mixed type 1/type2 [137,138]. ILC3s secrete IL-17, which leads to airway mucosal neutrophilia in some forms of asthma and CRS. A randomized, placebo-controlled double-blind trial was performed in 300 patients with moderate to severe asthma by using anti-IL-17, brodalumab, and it did not show a remarkable effect [139].

#### 6.4.3. Anti-IL-8

Neutrophils have surface IL-8 receptors and are the main target cells for IL-8 responses. Anti-IL-8R, CXCR2, has been shown to reduce airway neutrophilia [140]. Two placebo-controlled studies with CXCR2 antagonists have been performed in severe (neutrophilic) asthma patients [141,142]. The results did not show clinical effectiveness; however, in one of the studies, a reduction in sputum and blood neutrophils was observed [141].

#### 6.4.4. CRTH2 Antagonists

In addition to monoclonal antibodies, other molecules are also under investigation for future therapeutics of airway diseases. An example of these are chemoattractant receptor-homologous molecule (CRTH2) antagonists. Prostaglandin D2 (PGD2) is an arachidonic acid metabolite of the cyclooxygenase (COX) pathway. It plays a role in the pathophysiology of allergic rhinitis, CRS, and asthma [143]. PGD2 acts via DP1 and DP2 receptors, and CRTH2. PGD2 links adaptive and innate immune pathways via DP2 receptors located on Th2 cells, ILC2s, and eosinophils. Hence, PGD2 might be a good target for type 2 disorders [144,145]. CRTH2 antagonists represent a category of small molecules that have been discovered to have therapeutic potential for asthma [146,147]. CRTH2 antagonists have decreased the allergen-mediated airway responses of the upper [148] and lower airways [149,150]. CRTH2 antagonist has been given as monotherapy or in combination with standard therapy to patients with mild to moderate asthma, and it has shown a modest effectiveness on symptom scores, disease control, lung function, and inflammatory markers [151,152,153,154]. CRTH2 antagonists are proposed to have therapeutic effectiveness similar to antihistamines [148] and leukotriene receptor antagonists [5,154]. Dual DP/CRTH2 antagonist (AMG853) treatment for 12 weeks failed to show clinical effectiveness in patients with moderate to severe asthma [155]. Another CRTH2 receptor antagonist, fevipiprant, for 12 weeks, has shown to improve clinical and physiological parameters and to reduce airway eosinophils in patients with moderate-severe asthma [154], and it reduced asthma exacerbations moderately, but not significantly, in 52-week phase 3 trials in patients with severe asthma [156].

## 7. Conclusions

Taken together, monoclonal antibodies have several physiological and pathomechanistic roles in asthma, allergic rhinitis, and chronic rhinosinusitis. Local IgE production has been mostly studied. Future studies of other antibodies and their role in the pathomechanisms of inflammatory airway diseases are needed. Several monoclonal antibody treatments have indication for severe type 2 asthma; these are anti-IgE omalizumab, anti-IL-5 mepolizumab/reslizumab, anti-IL-5R benralizumab and anti-IL-4Ralpha dupilumab. Studies show that these treatments have an effect in patients with co-morbid severe type 2 asthma and CRS. For the treatment of severe CRSwNP, dupilumab and omalizumab are currently approved, and more are probably to come in the future. In addition to ongoing trials of the above-mentioned monoclonal antibodies, several other monoclonal antibodies are under active investigation. In addition to type 2 diseases, there is a high need to investigate therapeutic targets also for type 2 low asthma and CRS.

## Figures and Tables

**Figure 1 ijms-21-09477-f001:**
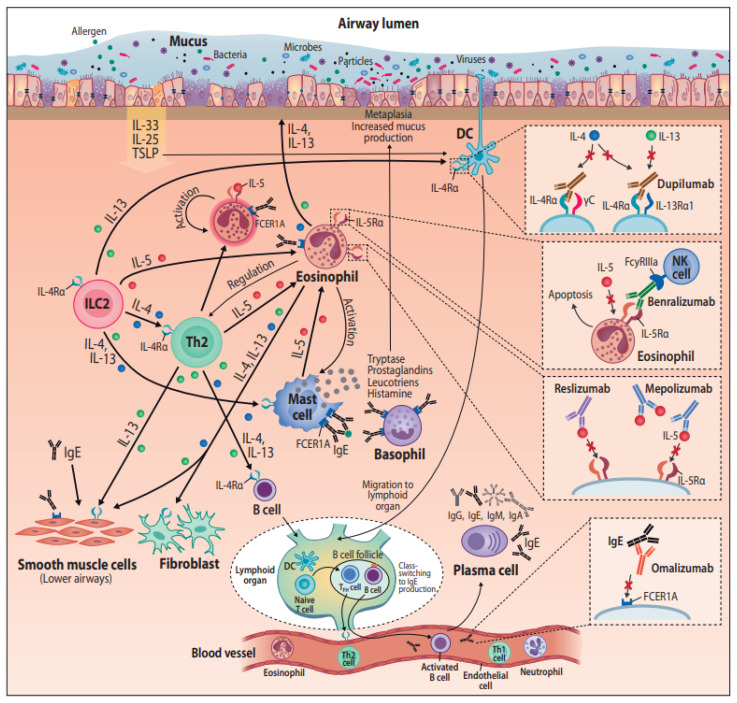
Monoclonal antibodies in the treatment of airway diseases, with their postulated pathways. Abbreviations: DC = dendritic cell, FCER1A = Fc fragment of Immunoglobulin E receptor 1A, FcyRIIIa = Fc fragment of IgG low affinity IIIa receptor, IgA = Immunoglobulin A, IgE = Immunoglobulin E, IgG = Immunoglobulin G, IgM = Immunoglobulin M, IL(-4, -4Rα, -5, -5Rα, -13, -13Rα, -25, -33) = Interleukin(-type), ILC2 = Group 2 innate lymphoid cells, NK cell = Natural killer cell, TFH cell = T follicular helper cell, Th1 = T helper type 1, Th2 = T helper type 2, TSLP = Thymic stromal lymphopoietin.

**Table 1 ijms-21-09477-t001:** Monoclonal antibodies in the treatment of different airway diseases.

Therapy (Target)	Asthma	CRSwNP	Dose	Response
Omalizumab (anti-IgE)	Severe allergic asthma with perennial allergy	Approved as add-on therapy for adults with severe CRSwNP by EMA in August 2020	According to weight and total S-IgE value, every 2–4 weeks s.c.	Asthma: Reduction in exacerbations, improvement in symptoms, asthma related QoL↑, FEV1↑
		Pending FDA approval [41]		CRSwNP: Reduction in symptom score and nasal polyp score
Mepolizumab (anti-IL5)	Severe eosinophilic asthma with B-eos>300cells/ul	Ongoing studies for use in CRSwNP [41]	100 or 300 mg s.c. every four weeks	Asthma: Reduction in exacerbations, improvement in symptoms, B-eos ↓, asthma related QoL↑, FEV1↑
Reslizumab (anti-IL5)	Severe eosinophilic asthma with B-eos>400cells/ul	Ongoing studies	According to weight every four weeks i.v.	Asthma: Reduction in exacerbations, improvement in symptoms, B-eos↓, Asthma related QoL↑, FEV1↑
Benralizumab (anti-IL5R)	Severe eosinophilic asthma with B-eos>300cells/ul	Ongoing studies for use in CRSwNP [41]	30 mg every 4 weeks s.c. three times and then 30 mg every 8 weeks s.c.	Asthma: Reduction in exacerbations, improvement in symptoms, B-eos↓
Dupilumab (anti-IL4Ralpha)	Severe eosinophilic asthma with B-eos>300cells/ul	Severe CRSwNP (approved by EMA and FDA)	First dose of 400 mg/600 mg s.c. according to weight, then 200 mg/300 mg every 2 weeks s.c.	Asthma: Reduction in exacerbations, improvement in symptoms, B-eos ↓, Asthma related QoL↑, FEV1↑
				CRSwNP: polyp size reduction, reduction in OCS and surgeries, improvement in symptom score

Abbreviations: B-eos = blood eosinophils, CRSwNP = chronic rhinosinusitis with nasal polyps, EMA = European Medicines Agency, FDA = U.S. Food and Drug Administration, FEV1 = forced expiratory volume in one second, i.v. = intravenous, NO = nitric oxide, OCS = oral corticosteroids, s.c. = subcutaneous, QoL = Quality of Life.

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
