# Peer review of "Monoclonal Antibodies and Airway Diseases"

_ijms, 2020, doi:10.3390/ijms21249477_

Round 1

Reviewer 1 Report

The authors review the role of monoclona antibodies in the treagment of asthma and chronic rhinosinusitis.

COMMENTS

Lines  87-88 Typical asthma symptoms include recurrent or prolonged (over 8 weeks) cough, wheezing, dyspnea, nighttime symptoms and mucus production.

Producing mucus is a physiological function of the airways, what happens in asthma is an overproduction or hyperproduction of mucus

Lines 88-89 Reversible airway obstruction can be diagnosed by spirometry, PEF monitoring, methacholine or mannitol challenge or by exercise test.

With methacholine, mannitol and exercise tests, bronchial hyperresponsiveness is detected rather than bronchoversibility

Lines 101-103 CRS with nasal polyps (CRSwNP) (2-4%) and without nasal polyps (CRSsNP) (10-20%), diagnosed after endoscopic evaluation of the presence of bilateral polyps in the middle meatus.

2-4% added to 10-20% give a maximum of 24% classified. Where are the remaining 76% of CRS patients classified?

Lines 139-140 . ASA desensitization has also been used to treat this 139 group of patients , unfortunately not all patients benefit from it . Fortunately, monoclonal 140 antibodies seem to provide help for this group of patients.

Desensitization is rarely used in Europe to treat CRS patients. For this reason, both the few patients with CRS in whom desensitizing treatment has failed and the majority of patients with severe CRS who have not been desensitized are fortunate to have a new and effective therapy for their disease.

Lines 193-194 Monoclonal anti-virus antibodies play a key role in antiviral response.

You mean monoclonal antibodies with antiviral effects. Actually monoclonal antibodies are not directed against viruses, the anti-virus effect is indirect, for example by improving the antiviral capacity of dendritic cells.

Line 304. Monoclonal antibodies in asthma treatment.

Recent studies carried out by M Taguinuchi's group have shown that omalizumab is very effective in the treatment of patients with NERD, to the point that in a high percentage of patients it manages to reverse hypersensitivity to NSAIDs. It is worth commenting for its singularity.

Author Response

Point-by-point response

We would like to thank the Editor and Reviewers for their excellent comments. We have now revised the manuscript and language as amended. When revising the text, we noticed that some literature references were missing. We apologize for this and have now added the lacking references in the text, the new references are listed below.

Reviewer 1

The authors review the role of monoclona antibodies in the treagment of asthma and chronic rhinosinusitis.

COMMENTS

Lines  87-88 Typical asthma symptoms include recurrent or prolonged (over 8 weeks) cough, wheezing, dyspnea, nighttime symptoms and mucus production.

Producing mucus is a physiological function of the airways, what happens in asthma is an overproduction or hyperproduction of mucus

Response: We agree with the reviewer and have clarified the sentence as follows: “Typical asthma symptoms include recurrent or prolonged (over 8 weeks) cough, wheezing, dyspnea, nighttime symptoms and overproduction of mucus.”

Lines 88-89 Reversible airway obstruction can be diagnosed by spirometry, PEF monitoring, methacholine or mannitol challenge or by exercise test.

With methacholine, mannitol and exercise tests, bronchial hyperresponsiveness is detected rather than bronchoversibility

Response: Thank you for the comment. We agree and have clarified the sentence as follows: “Reversible airway obstruction can be diagnosed by spirometry and PEF monitoring, and bronchial hyperresponsiveness is detected with methacholine or mannitol challenge or by exercise test.”

Lines 101-103 CRS with nasal polyps (CRSwNP) (2-4%) and without nasal polyps (CRSsNP) (10-20%), diagnosed after endoscopic evaluation of the presence of bilateral polyps in the middle meatus.

2-4% added to 10-20% give a maximum of 24% classified. Where are the remaining 76% of CRS patients classified?

Response: Thank you for this comment. We have clarified the paragraph as follows: The overall prevalence of CRS has been estimated to be 10.9%, with wide variation between countries (6.9% to 27.1%) (ref. Hastan D. et al.). Traditionally, CRS has been classified into two subtypes: CRS with nasal polyps (CRSwNP) and without nasal polyps (CRSsNP), diagnosed after endoscopic evaluation of the presence of bilateral polyps in the middle meatus. Data on the overall prevalence of CRSwNP is limited, but it is estimated to be approximately 2-3% (ref. Larsen et al., Johansson et al., EPOS 2020).

Lines 139-140 . ASA desensitization has also been used to treat this 139 group of patients , unfortunately not all patients benefit from it . Fortunately, monoclonal 140 antibodies seem to provide help for this group of patients.

Desensitization is rarely used in Europe to treat CRS patients. For this reason, both the few patients with CRS in whom desensitizing treatment has failed and the majority of patients with severe CRS who have not been desensitized are fortunate to have a new and effective therapy for their disease.

Response: We agree and have added discussion of this. ASA desensitization (ATAD) is an important therapeutical aspect for patients with uncontrolled NERD+CRSwNP. We have thus added information of ATAD and have added discussion of the role of biologics in this patient group. We have modified the paragraph accordingly:

Oral ASA treatment after desensitization (ATAD) has shown to be effective in improving QOL and total nasal symptom score in patients with N-ERD (ref. EPOS 2020 and Stevenson et al.). However, the treatment is associated with adverse effects (typically gastrointestinal) and should be continued strictly on a daily basis (ref. EPOS 2020). Studies with ATAD show high discontinuation rates and not all patients benefit from it (ref. EPOS 2020 and Laulajainen-Hongisto et al.). Monoclonal antibodies have shown efficacy in patients with severe CRS + N-ERD.

Lines 193-194 Monoclonal anti-virus antibodies play a key role in antiviral response.

You mean monoclonal antibodies with antiviral effects. Actually monoclonal antibodies are not directed against viruses, the anti-virus effect is indirect, for example by improving the antiviral capacity of dendritic cells.

Response: Thank you for this important comment. We agree that the anti-virus effects of monoclonal antibodies are indirect, for example by improving antiviral capacity of dendritic cells. Antiviral mAb therapy usually targets directly and rapidly the infectious agent, yet evidence has revealed that antiviral mAbs may be used to recruit the endogenous immune systems of infected organisms to induce long-lasting vaccine-like effects. We have now clarified this in the text (page 6) as follows:

“The anti-virus effects of monoclonal antibodies are indirect, for example by improving antiviral capacity of dendritic cells (Yu et al. 2020). Antiviral mAb therapy usually targets directly and rapidly the infectious agent, yet evidence has revealed that antiviral mAbs may be used to recruit the endogenous immune systems of infected organisms to induce long-lasting vaccine-like effects (Pelegrin et al 2015).”

Line 304. Monoclonal antibodies in asthma treatment.

Recent studies carried out by M Taguinuchi's group have shown that omalizumab is very effective in the treatment of patients with NERD, to the point that in a high percentage of patients it manages to reverse hypersensitivity to NSAIDs. It is worth commenting for its singularity.

Response: We apologize to have left this important aspect. We have now added discussion on the effect of Omalizumab in patients with NSAIDs-exacerbated respiratory disease (page 10) (Förster-Ruhrmann et al. 2020).

Added references:

Toskala, E.; Kennedy, D.W. Asthma risk factors. Int. Forum Allergy Rhinol. 2015, 5, S11–S16, doi:10.1002/alr.21557.

Bousquet, J.; Anto, J.M.; Bachert, C.; Baiardini, I.; Bosnic-Anticevich, S.; Canonica, G.W.; Melen, E.; Palomares, O.; Scadding, G.K.; Togias, A.; et al. Allergic rhinitis. Nat. Rev. 2020, in press.

Barnes, P.J. Intrinsic asthma: not so different from allergic asthma but driven by superantigens? Clin. Exp. Allergy 2009, 39, 1145–1151, doi:10.1111/j.1365-2222.2009.03298.x.

Panda, S.; Ding, J.L. Natural Antibodies Bridge Innate and Adaptive Immunity. J. Immunol. 2015, 194, 13–20, doi:10.4049/jimmunol.1400844.

Marshall, J.S.; Warrington, R.; Watson, W.; Kim, H.L. An introduction to immunology and immunopathology. Allergy, Asthma Clin. Immunol. 2018, 14, 49.

Wang, L.L.; Moshiri, A.S.; Novoa, R.; Simpson, C.L.; Takeshita, J.; Payne, A.S.; Chu, E.Y. Comparison of C3d immunohistochemical staining to enzyme-linked immunosorbent assay and immunofluorescence for diagnosis of bullous pemphigoid. J. Am. Acad. Dermatol. 2020, 83, 172–178, doi:10.1016/j.jaad.2020.02.020.

Black, C.A. A brief history of the discovery of the immunoglobulins and the origin of the modern immunoglobulin nomenclature. Immunol. Cell Biol. 1997, 75, 65–68.

Platts-Mills, T.A.E.; Heymann, P.W.; Commins, S.P.; Woodfolk, J.A. The discovery of IgE 50 years later. Ann. Allergy, Asthma Immunol. 2016, 116, 179–182, doi:10.1016/j.anai.2016.01.003.

Larsen, K.; Tos, M. The Estimated Incidence of Symptomatic Nasal Polyps. http://dx.doi.org/10.1080/00016480252814199 2009, doi:10.1080/00016480252814199.

Johansson, L.; Åkerlund, A.; Holmberg, K.; Melén, I.; Bende, M. Prevalence of nasal polyps in adults: The Skövde population-based study. Ann. Otol. Rhinol. Laryngol. 2003, 112, 625–629, doi:10.1177/000348940311200709.

Gough, H.; Grabenhenrich, L.; Reich, A.; Eckers, N.; Nitsche, O.; Schramm, D.; Beschorner, J.; Hoffmann, U.; Schuster, A.; Bauer, C.P.; et al. Allergic multimorbidity of asthma, rhinitis and eczema over 20 years in the German birth cohort MAS. Pediatr. Allergy Immunol. 2015, 26, 431–437, doi:10.1111/pai.12410.

Toppila-Salmi, S.; Chanoine, S.; Karjalainen, J.; Pekkanen, J.; Bousquet, J.; Siroux, V. Risk of adult-onset asthma increases with the number of allergic multimorbidities and decreases with age. Allergy Eur. J. Allergy Clin. Immunol. 2019, 74, 2406–2416, doi:10.1111/all.13971.

Bachert, C.; Marple, B.; Hosemann, W.; Cavaliere, C.; Wen, W.; Zhang, N. Endotypes of Chronic Rhinosinusitis with Nasal Polyps: Pathology and Possible Therapeutic Implications. J. Allergy Clin. Immunol. Pract. 2020, 8, 1514–1519, doi:10.1016/j.jaip.2020.03.007.

Yu, X.; Cragg, M.S. Engineered antibodies to combat viral threats. Nature 2020, doi:10.1038/d41586-020-03196-2.

Pelegrin, M.; Naranjo-Gomez, M.; Piechaczyk, M. Antiviral Monoclonal Antibodies: Can They Be More Than Simple Neutralizing Agents? Trends Microbiol. 2015, 23, 653–665, doi:10.1016/j.tim.2015.07.005.

Cardell, L.O.; Stjärne, P.; Jonstam, K.; Bachert, C. Endotypes of chronic rhinosinusitis: Impact on management. J. Allergy Clin. Immunol. 2020, 145, 752–756, doi:10.1016/j.jaci.2020.01.019.

Manka, L.A.; Wechsler, M.E. Selecting the right biologic for your patients with severe asthma. Ann. Allergy, Asthma Immunol. 2018, 121, 406–413.

Förster-Ruhrmann, U.; Stergioudi, D.; Pierchalla, G.; Fluhr, J.W.; Bergmann, K.C.; Olze, H. Omalizumab in patients with NSAIDs-exacerbated respiratory disease. Rhinology 2020, 58, 226–232, doi:10.4193/Rhin19.318.

Reviewer 2 Report

The article summarizes monoclonal antibodies role in treatment of asthma, CRS and allergic rhinitis. It is well written and gives opportunity to summarize the knowledge and future developments of new therapeuticals. First part, about CRS, AR, asthma could be abbreviated, but altogether nicely written. 

Author Response

We would like to thank the Editor and Reviewers for their excellent comments. We have now revised the manuscript and language as amended. When revising the text, we noticed that some literature references were missing. We apologize for this and have now added the lacking references in the text, the new references are listed below.

Added references:

Toskala, E.; Kennedy, D.W. Asthma risk factors. Int. Forum Allergy Rhinol. 2015, 5, S11–S16, doi:10.1002/alr.21557.

Bousquet, J.; Anto, J.M.; Bachert, C.; Baiardini, I.; Bosnic-Anticevich, S.; Canonica, G.W.; Melen, E.; Palomares, O.; Scadding, G.K.; Togias, A.; et al. Allergic rhinitis. Nat. Rev. 2020, in press.

Barnes, P.J. Intrinsic asthma: not so different from allergic asthma but driven by superantigens? Clin. Exp. Allergy 2009, 39, 1145–1151, doi:10.1111/j.1365-2222.2009.03298.x.

Panda, S.; Ding, J.L. Natural Antibodies Bridge Innate and Adaptive Immunity. J. Immunol. 2015, 194, 13–20, doi:10.4049/jimmunol.1400844.

Marshall, J.S.; Warrington, R.; Watson, W.; Kim, H.L. An introduction to immunology and immunopathology. Allergy, Asthma Clin. Immunol. 2018, 14, 49.

Wang, L.L.; Moshiri, A.S.; Novoa, R.; Simpson, C.L.; Takeshita, J.; Payne, A.S.; Chu, E.Y. Comparison of C3d immunohistochemical staining to enzyme-linked immunosorbent assay and immunofluorescence for diagnosis of bullous pemphigoid. J. Am. Acad. Dermatol. 2020, 83, 172–178, doi:10.1016/j.jaad.2020.02.020.

Black, C.A. A brief history of the discovery of the immunoglobulins and the origin of the modern immunoglobulin nomenclature. Immunol. Cell Biol. 1997, 75, 65–68.

Platts-Mills, T.A.E.; Heymann, P.W.; Commins, S.P.; Woodfolk, J.A. The discovery of IgE 50 years later. Ann. Allergy, Asthma Immunol. 2016, 116, 179–182, doi:10.1016/j.anai.2016.01.003.

Larsen, K.; Tos, M. The Estimated Incidence of Symptomatic Nasal Polyps. http://dx.doi.org/10.1080/00016480252814199 2009, doi:10.1080/00016480252814199.

Johansson, L.; Åkerlund, A.; Holmberg, K.; Melén, I.; Bende, M. Prevalence of nasal polyps in adults: The Skövde population-based study. Ann. Otol. Rhinol. Laryngol. 2003, 112, 625–629, doi:10.1177/000348940311200709.

Gough, H.; Grabenhenrich, L.; Reich, A.; Eckers, N.; Nitsche, O.; Schramm, D.; Beschorner, J.; Hoffmann, U.; Schuster, A.; Bauer, C.P.; et al. Allergic multimorbidity of asthma, rhinitis and eczema over 20 years in the German birth cohort MAS. Pediatr. Allergy Immunol. 2015, 26, 431–437, doi:10.1111/pai.12410.

Toppila-Salmi, S.; Chanoine, S.; Karjalainen, J.; Pekkanen, J.; Bousquet, J.; Siroux, V. Risk of adult-onset asthma increases with the number of allergic multimorbidities and decreases with age. Allergy Eur. J. Allergy Clin. Immunol. 2019, 74, 2406–2416, doi:10.1111/all.13971.

Bachert, C.; Marple, B.; Hosemann, W.; Cavaliere, C.; Wen, W.; Zhang, N. Endotypes of Chronic Rhinosinusitis with Nasal Polyps: Pathology and Possible Therapeutic Implications. J. Allergy Clin. Immunol. Pract. 2020, 8, 1514–1519, doi:10.1016/j.jaip.2020.03.007.

Yu, X.; Cragg, M.S. Engineered antibodies to combat viral threats. Nature 2020, doi:10.1038/d41586-020-03196-2.

Pelegrin, M.; Naranjo-Gomez, M.; Piechaczyk, M. Antiviral Monoclonal Antibodies: Can They Be More Than Simple Neutralizing Agents? Trends Microbiol. 2015, 23, 653–665, doi:10.1016/j.tim.2015.07.005.

Cardell, L.O.; Stjärne, P.; Jonstam, K.; Bachert, C. Endotypes of chronic rhinosinusitis: Impact on management. J. Allergy Clin. Immunol. 2020, 145, 752–756, doi:10.1016/j.jaci.2020.01.019.

Manka, L.A.; Wechsler, M.E. Selecting the right biologic for your patients with severe asthma. Ann. Allergy, Asthma Immunol. 2018, 121, 406–413.

Förster-Ruhrmann, U.; Stergioudi, D.; Pierchalla, G.; Fluhr, J.W.; Bergmann, K.C.; Olze, H. Omalizumab in patients with NSAIDs-exacerbated respiratory disease. Rhinology 2020, 58, 226–232, doi:10.4193/Rhin19.318.